# Understanding the epidemiology and pathogenesis of *Mycobacterium tuberculosis* with non-redundant pangenome of epidemic strains in China

Yang Zhou[1,2], Richard Anthony[3], Shengfen Wang[1], Hui Xia[1], Xichao Ou[1], Bing Zhao[1], Yuanyuan Song[1], Yang Zheng[1], Ping He[1], Dongxin Liu[1], Yanlin Zhao[1]*, Dick van Soolingen[2]*

1 National Center for Tuberculosis Control and Prevention, Chinese Center for Disease Control and Prevention, Changping District, Beijing, China, 2 Radboudumc Research Institute, Radboud University, Houtlaan XZ, Nijmegen, The Netherlands, 3 National Tuberculosis Reference Laboratory, Centre for Infectious Disease Control, National Institute for Public Health and the Environment (RIVM), Bilthoven, The Netherlands.

* zhaoyl@chinacdc.cn (YZ); dick.van.soolingen@ziggo.nl (DS)

## Abstract

Tuberculosis is a major public health threat resulting in more than one million lives lost every year. Many challenges exist to defeat this deadly infectious disease which address the importance of a thorough understanding of the biology of the causative agent *Mycobacterium tuberculosis* (MTB). We generated a non-redundant pangenome of 420 epidemic MTB strains from China including 344 Lineage 2 strains, 69 Lineage 4 strains, six Lineage 3 strains, and one Lineage 1 strain. We estimate that MTB strains have a pangenome of 4,278 genes encoding 4,183 proteins, of which 3,438 are core genes. However, due to 99,694 interruptions in 2,447 coding genes, we can only confidently confirm 1,651 of these genes are translated in all samples. Of these interruptions, 67,315 (67.52%) could be classified by various genetic variations detected by currently available tools, and more than half of them are due to structural variations, mostly small indels. Assuming a proportion of these interruptions are artifacts, the number of active core genes would still be much lower than 3,438. We further described differential evolutionary patterns of genes under the influences of selective pressure, population structure and purifying selection. While selective pressure is ubiquitous among these coding genes, evolutionary adaptations are concentrated in 1,310 genes. Genes involved in cell wall biogenesis are under the strongest selective pressure, while the biological process of disruption of host organelles indicates the direction of the most intensive positive selection. This study provides a comprehensive view on the genetic diversity and evolutionary patterns of coding genes in MTB which may deepen our understanding of its epidemiology and pathogenicity.

**Data availability statement:** All sequencing data is available in project PRJNA573798 on NCBI server.

**Funding:** Authors received the funding Ou Xichao & Zhao Yanlin Grant numbers

Zhao Yanlin : 2022YRC2305203 Ou Xichao : 2022YRC2305204 Project Title & Grand Numbers: Establishment and application of China's AIDS and tuberculosis pathogen gene database and intelligent precision prevention and control platform (2022YRC2305200) Sub-project Title & Grand Numbers : Creation of a national representative tuberculosis pathogen genetic sequence database and its analysis tools (2022YRC2305203) Sub-project Title & Grand Numbers : Research and application of tuberculosis transmission network and its molecular analysis tools in China (2022YRC2305204) Full name of funder 2022YRC2305203 & 2022YRC2305204 : Ministry of Science and Technology of the People's Republic of China Url of funder Ministry of Science and Technology of the People's Republic of China: https://www.most.gov.cn/index.html The funder had no role in study design, data collection and analysis, decision to publish, or preparation of the manuscript.

**Competing interests:** The authors have declared that no competing interests exist.

## Introduction

Tuberculosis (TB) took 1.3 million lives in 2022 globally, making it the second deadliest infectious disease just after COVID-19.[1] The World Health Organization (WHO) has set the goal to eliminate tuberculosis as a public health problem by 2035, which requires reducing TB incidence by 90% and mortality by 95% compared to 2015. New vaccines, better drugs and improved diagnostic tools are key advances needed to end the TB pandemic. However, only one now old vaccine with limited efficacy, the so-called BCG developed from attenuated *M. bovis*, has been licensed for TB. Although recently new drugs, including bedaquiline, pretomanid and delamanid, have been added to the list of drugs available to treat TB in the past decades, strains extremely resistant even to these new drugs have already been observed. This stunted progress is partially due to the lack of incentives and underfunding, but also underlines our lack of understanding of the biological process and mechanisms relating to issues such as host-pathogen interaction and drug resistance development.

The first whole genome sequence of MTB (*Mycobacterium tuberculosis*) strain H37Rv was published in 1998 and has been used as the backbone for mapping short reads from high-throughput sequencing platforms to identify genetic variations, primarily single nucleotide polymorphisms (SNPs) and small indels described in most studies.[2] This information can be used to infer the phylogeny and evolution of epidemic strains, to monitor recent transmission and outbreaks, and to predict drug resistance phenotypes.[3–6] These milestone findings suggest that SNPs and small indels indeed correspond to a large proportion of the observed diversities in clinical MTB isolates. However, the limitation of this pipeline is also obvious since genetic variations in the regions not presented in the reference genome are not included. Comparative genome analysis has already shown that large sequence polymorphisms (LSPs) between strains are not only related to the genealogical delineation, such as Lineage 2 of MTB defined by RD105, but are also associated with important biological processes.[7,8] For example, two genes *mmpS6* and *mmpL6* are both deleted or truncated in "modern" MTB strains due to the deletion of TbD1 region, which may facilitate the successful expansion of modern strains globally through a hypoxia-induced copper response and increased virulence.[9,10] Another region of difference, RD1, harbors several open reading frames (ORFs) encoding ESX-1 secretion system proteins including CFP-10 and ESAT-6 which are crucial to virulence and phagosomal escape.[11,12] The RD1 deletion in *M. bovis* is related to the attenuation of the BCG vaccine.[13] Besides these LSPs which result in complete deletions of genes, other types of structural variations (SVs), such as IS*6110* insertions in coding sequences, partial deletions of genes by large deletions (>100 bp) or interruptions by large insertions, are also missed by the standard pipeline and may have important consequences related to important biological processes.[12,14,15] By focusing only on SNPs and small indels, it's assumed that at least a certain proportion of genetic diversity information in clinical strains of MTB is missed, thus we propose a more comprehensive approach to study genomic data.

Accumulation of high through-put sequencing data has allowed the development of a new study field: pangenome analysis.[16,17] The term pangenome refers to all the genetic content of a given dataset, usually a species. The shared genomic content or genes in all genomes are referred to as core genes; other genes presented in several but not all genomes are called dispensable or accessory genes which may be introduced by horizontal gene transfer, gene duplication/deletion or other SVs. Accessory genes are supposed to account for much of the diversity among clinical strains of microorganisms.[18] This is especially true for species with a large number of accessory genes, such as *E. coli*, which has high genome plasticity and a very large gene pool.[19] MTB is known to have a conserved genome with no plasmids detected and horizontal gene transfer is very rare in modern strains if it exists at all. Thus, it is questionable how much of the diversity observed in clinical practice could be explained by the presence/absence of accessory genes. The drawback of the presence/absence approach is that the variations within genes are ignored, which might contribute to the genetic diversity at different levels. Thus, for a microorganism with a conserved genome such as MTB, a combined approach by integrating the pangenome and within gene variation information is desirable.

In this study, we developed a pipeline to generate a non-redundant pangenome of MTB addressing the problems outlined above from the available short read sequencing data.[20] Based on the pangenome matrix and variations detected with dedicated tools, we described the genetic variation at nucleotide, gene and protein levels. Based on this comprehensive profile of genetic diversity in the pangenome, we characterized the evolutionary patterns of all coding genes under the influence of selective pressure, population structure and purifying selection, which reflects the complex interactions between these factors. Gene ontology analysis of genes showing different evolutionary patterns has revealed some primary characteristics of the MTB biology and the interactions between MTB and the environment.

## Method

### Sampling and WGS

The 420 clinical MTB isolates used in this retrospective study were selected from the collection of the first national drug resistance baseline survey in China from April 1 to December 31, 2007.[21] Written informed consent was obtained from each participant. All participants were anonymized by uniformed participants identifier. Drug susceptibility testing was performed for six drugs including isoniazid, rifampin, ofloxacin, streptomycin, ethambutol and kanamycin.[21] Selection criteria were based on drug resistance patterns, geographic location of isolation, and genotypes such as spoligotype and MIRU-VNTR profile, in order to represent the diversity in different geographic regions and the population structure of the epidemic strains. The phylogenetic and evolutionary analysis based on whole genome sequencing were published previously by our laboratory and all sequencing data is available in project **PRJNA573798 (**https://www.ncbi.nlm.nih.gov/bioproject/?term=PRJNA573798**)** on the NCBI server.[20] Briefly, all samples were sequenced on an Illumina platform using single end or paired end library. After filtering out repeat regions, PE/PPE_PGRS genes and insertion sequences, phylogenetic trees were constructed with MEGA using a maximum likelihood algorithm.[22] For the following analysis, the reference genome of H37Rv (NC_000962.3) was used for annotation and variation detection. This study has been approved within the project (2022YRC2305200) by the Ethics Review Board of China CDC (No. 202223) and the data was accessed for the pangenome analysis in November 30, 2022.

### Assembly and annotation

The same sequencing data was used to produce the assembly of each strain using the program Spades with k-mer size auto-detected and coverage cut-off set to off and trying to reduce the number of mismatches and short indels.[23] The resulting contigs were first blasted against the NCBI nucleotide database to remove possible contamination from the previous steps. All blast hits of e-value > 1e-5 were excluded. Preserved contigs were excluded if aligned to species other than MTBC. The quality of assemblies was evaluated with QUAST.[24] All contigs longer than 200 bp that passed the

contamination filtering were annotated by Prokka [25] to predict coding sequences (CDSs) with default parameters. The produced gbk file was used to search for homologues genes.

## Constructing non-redundant pangenome

Two approaches are combined to generate the full-set of non-redundant pangenome: CDS prediction and blastn. CDSs predicted by Prokka were clustered by GET_HOMOLOGUES using COG algorithm.[26] When searching for clusters, the max e-value for blast was set to 0.01, the max intergenic size was set to 1,000 bp, minimum coverage was using the default 75%. At least one sequence was required for a cluster to be called. Only CDSs were considered for clustering. Clusters which were clustered or overlapped with Rv by>=60 bp were considered as redundant to Rv and were removed from candidate new genes list.[27] The left candidate new-/non-Rv genes went through self-overlapping filter by length to keep the longer CDSs if the overlap exceeds 60 bp.

Because Prokka can only predict valid ORFs, for those with interruptions such as frameshift mutations or other structure mutations, these genes are not predicted in the output or predicted with alternative ORFs, but could provide information on phylogeny and evolution. To further detect this part of information missed by prediction software, we applied *blastn* using consensus sequences of each cluster as the query sequence. Consensus sequences of the new candidate genes and sequences of all 3,906 CDSs and 112 other genetic features from H37Rv genome, including pseudogenes (30), tRNAs (45), ncRNA (20), rRNA (3), misc RNA (2), fragment of putative small regulatory RNA (10), and two other feature genes, were blasted to all contigs of 420 samples and the H37Rv genome. Blast hits with e-value < 1e-5, identity > 90% and query coverage > 30% (for Rv genes)/ > 90% (new candidate genes) were kept. *Blastn* results went through redundant checking using synteny and overlapping information. When two blast hits for the same gene in one sample have no overlap of their locations, they would be considered as gene duplications and only represented one homologues gene.

Finally, candidate new genes in complementary sequences to H37Rv genome were manually checked and filtered according to expert opinion by manual checking the similarity of complementary sequences and locations of the candidate new genes in the complementary sequences. All contigs were blasted to the H37Rv genome sequence and blast hits with e-value < 1e-5 and identity > 90% with the highest score were kept. Complementary sequences absent in the H37Rv genome were extracted based on blast results. All complementary sequences longer than 100 bp were compared to each other and those sharing more than 90% identity and more than 95% query coverage with e-value < 1e-5 were considered as homologous sequences. This step could filter alternative ORFs predicted in different samples that were not filtered out in previous steps. The whole pipeline is summarized in S1 Fig.

## Pangenome composition

The presence or absence of an individual gene was analyzed at different levels separately. At DNA level, if *blastn* identified a non-redundant best hit for a gene, it was considered as the presence of this gene. Conversion of gene sequence to protein sequence was done with Biopython package.[28] Because genetic variations such as SVs or non-sense point mutations could result in failure of *in silico* translation, only coding sequences that could be translated into protein sequences *in silico* were considered as valid ORFs. At protein level, only when a gene sequence had a valid ORF was it considered as present. Non-synonymous mutations in start/stop codon resulting in loss of start/stop codons, gain of an extra stop codon, frameshift mutations, and IS*6110* insertions in CDSs were considered as interruptions of translation. In addition, if > 90% of the gene sequence was deleted or not detected by *blastn*, it was considered as complete deletion.

We define core genes/proteins as genes/proteins presenting in all 420 strains and the reference genome, soft-core genes/proteins as genes/proteins which were deleted/interrupted in less than 5% (n = 21) of the whole sample population, shell genes/proteins as presenting in between 3 (inclusive) to 400 genomes, and cloud genes/proteins in less than 3 genomes.

## Detection of structural variations and high impact SNPs

Eleven open source software packages were used to call insertions and deletions, including assemblytics [29], bcftools [30], breakdancer [31], delly [32], lumpy-smoove [33], minimap2 [34], svaba [35], softsv [36], tiddit [37], unimap [38], and wham [39]. Called variants were first filtered based on the parameters with each method such as number of supporting reads. Inversions were not analyzed in this study. Deletions and insertions > 100 kb were also filtered without further check. Variants detected for each sample were merged by SURVIVOR only if supported by more than one method (for deletions > 1,000 bp, more than four methods were required) with restrain to the same variation type and breakpoints within 100 bp.[40] High impact SNPs were defined as SNPs that cause loss of the start/stop codon or gain of an extra stop codon within ORFs. The standard pipeline based on bwa and bcftools was used to detect the SNPs and the effect was predicted by snpEff.[41]

The pangenome matrix at gene level from composition analysis was compared with the results from SURVIVOR. Deletions >1,000 bp without corresponding gene deletion in the pangenome matrix were checked manually with IgV.[42] The gene deletions in the pangenome matrix that were not detected by the 11 methods were checked by viewing in IgV and added to the dataset if confirmed. IS*6110* insertions were detected in paired end samples with ISMapper.[43]

## Genetic diversity, selective pressures, positive selection and population stratification

Tajima's D [44] and nucleotide diversity π [45] for each coding gene were calculated using Dnasp (v6).[46] Interrupted gene copies were excluded before calculating the statistics under the assumption that these sequences evolve faster than valid ORFs. Significance for Tajima's D was given by the table in the original paper by Tajima.[44] Homoplasy was detected by homoplasyFinder using the default parameters.[47] Homoplasy in SNPs were only detected in valid CDSs. For each gene tree used to find homoplasy SNPs in the specific gene, the phylogeny tree produced previously was re-used and genomes with interrupted CDSs were removed using the python package *ete* because homoplasyFinder requires all genomes in the tree presenting sequences for analysis.[20,48] SVs resulting in interruptions of the same CDS in the same genome were considered as homoplasy ignoring the type and the position information. Interquartile range (IQR) for homoplasy events was calculated by substracting the 25th percentile from the 75th percentile. A high homoplasy level was defined as more than the sum of IQR and the 75th percentile. Fixation index Fst for SNPs was calculated using command basic.stats from R package Hierfstat according to the equation 7.38-7.43 in page 164–5 of Nei (1987).[49,50] All negative Fsts were considered as zero. Significance for positive Fsts were calculated by permutating all samples among all groups for 1,000 times and counting the number of Fsts greater than or equal to the observed Fst using the same R package Hierfstat.

## Gene ontology and overrepresentation analysis

Coding genes under different evolutionary patterns were subjected to gene over-representation analysis using enrichGO function in clusterProfiler (version 4.14.4) package.[51] The background annotation database was constructed from annotations from PANTHER server (accessed in January 27, 2025) using AnnotationForge package (version 1.48.0) and GO.db package (version 3.20).[52] Bonferroni correction and Fisher exact test were implemented to find over-represented GO items in each group as compared to the background annotation database. The minimum annotated gene set size for analysis was set to 3. P-value, adjust P-value and q-value were all set to 0.05 to filter insignificant results. Only the most specific GO items were retrieved.

## Results

### M. *tuberculosis* has a conserved and closed pangenome

The data sample used in this study has been analyzed in a previous study which described the phylogeny and evolution of the epidemic MTB strains in China.[20] This sample set contained 344 Beijing family strains, 14 Lineage 4.2 strains, 15

Lineage 4.4 strains, 40 Lineage 4.5 strains, 6 Lineage 3 strains and one Lineage 1 strain, which were representative of the epidemic MTB strains in China and these strains were also prevalent in other regions globally.

The number of coding genes annotated for each genome varied from 4,167–4,337 (median = 4,194), resulting in a total of 1,707,532 coding gene sequences by *de novo* annotation from 420 strains and the H37Rv genome. After pair-wise comparison and clustering, the initial pangenome set automatically predicted by software consisted of 10,097 coding gene clusters, which included 3,754 singleton clusters. More than two thirds (1,167,634; 68.38%) of the predicted coding gene sequences were identical to the blast results. Besides Rv gene sequences, 66,412 sequences in 436 clusters remained after removing redundant (overlapping >=60 bp with Rv genes) CDSs predicted by Prokka. To deal with alternative ORFs annotated in different genomes, another round of blast using these 436 clusters as query sequences was done. After blasting consensus sequences of these 436 clusters to all contigs and repeating the process of removing redundance, 288 clusters were left. At this step, one cluster was removed because there were only two sequences of very different lengths in this cluster and no consensus sequence could be produced.

Among the remaining 288 clusters, 20 clusters harbored repeat sequences or were in insertion sequences such as IS*6110* and were removed from the final pangenome. There were 175 clusters which had non-redundant homologues sequences in the H37Rv genome and were considered as new Rv genes. Among the other 93 clusters, 22 clusters were in previously reported LSPs, such as RvD1 (four genes), RvD2 (4), RvD5(4), and TbD1 (1), and some less known LSPs, such as the RvD4494 (inserted at 2,219,418, six genes). Another 40 clusters were in complementary sequences (sequences not present in the H37Rv genome) adjacent to PPE/PE_PGRS genes or IS*6110*/IS*1532* insertion sequences or both. Seven genes were in complementary sequences adjacent to other highly polymorphic regions, such as *Rv2082*. These 69 genes were not detected in H37Rv genomes and were considered as non-Rv genes. Another 16 genes were not in complementary sequences to H37Rv but were interrupted by various kinds of SVs in H37Rv, such as *15427_dxs* interrupted by IS*6110*–15 insertion. These 16 clusters were also considered as non-Rv genes, which resulted in a total of 85 non-Rv genes. The eight genes left were excluded from the pangenome because of overlapping >= 60 bp in H37Rv (though in some genomes < 60 bp) or being in low quality short contigs (about 300 bp). Finally, 260 new genes were added to the final pangenome in addition to the 4,018 genes/features in H37Rv genome, including 175 new Rv genes and 85 non-Rv genes.

The final non-redundant pangenome constituted 4,278 genes, 4,183 of which were protein coding genes. Besides the 3,906 CDSs and 175 new Rv genes in the H37Rv genome, another 17 pseudo Rv genes interrupted in H37Rv might produce translation products in other genomes, so the total coding genes in H37Rv were 4,098 (Fig 1).

## Genetic variations in the pangenome

Among the 4,183 protein coding genes in this pangenome, 3,438 genes presented in all genomes, but only 1,651 genes had valid ORFs in all 421 genomes. We combined pangenome matrix with presence/absence information and genetic variations detected to see how the presence or absence at gene level and at protein level in the pangenome could be categorized by different genetic variations. Because most of the available tools to detect SVs and SNPs with short reads sequencing data require a linear reference genome, genetic variations such as SNPs and SVs were only detected for the 4,098 coding genes (including pseudo-Rv genes) in H37Rv genome. We also included insertions of IS*6110* to categorize the interruptions.

With dedicated tools, we detected 76,671 SVs (7,334 unique SVs) in the 4,098 coding genes in total, including 49,655 deletions, 26,700 insertions and 316 inversions. Inversions were not included in the following analysis. For the 325 paired-end samples, 1,860 IS*6110* insertions at 411 positions in 311 samples were detected. Using the standard pipeline, there were 16,151 high impact point mutations in start/stop codons, such as those causing loss of a start/stop codon or gain of an extra stop codon, detected in 1,007 genes. Some gene copies contained multiple types of variations which were referred to as "complex" in this study. A "complex SV" composed of "indels" plus an "IS*6110*" insertion was detected 35 times in our

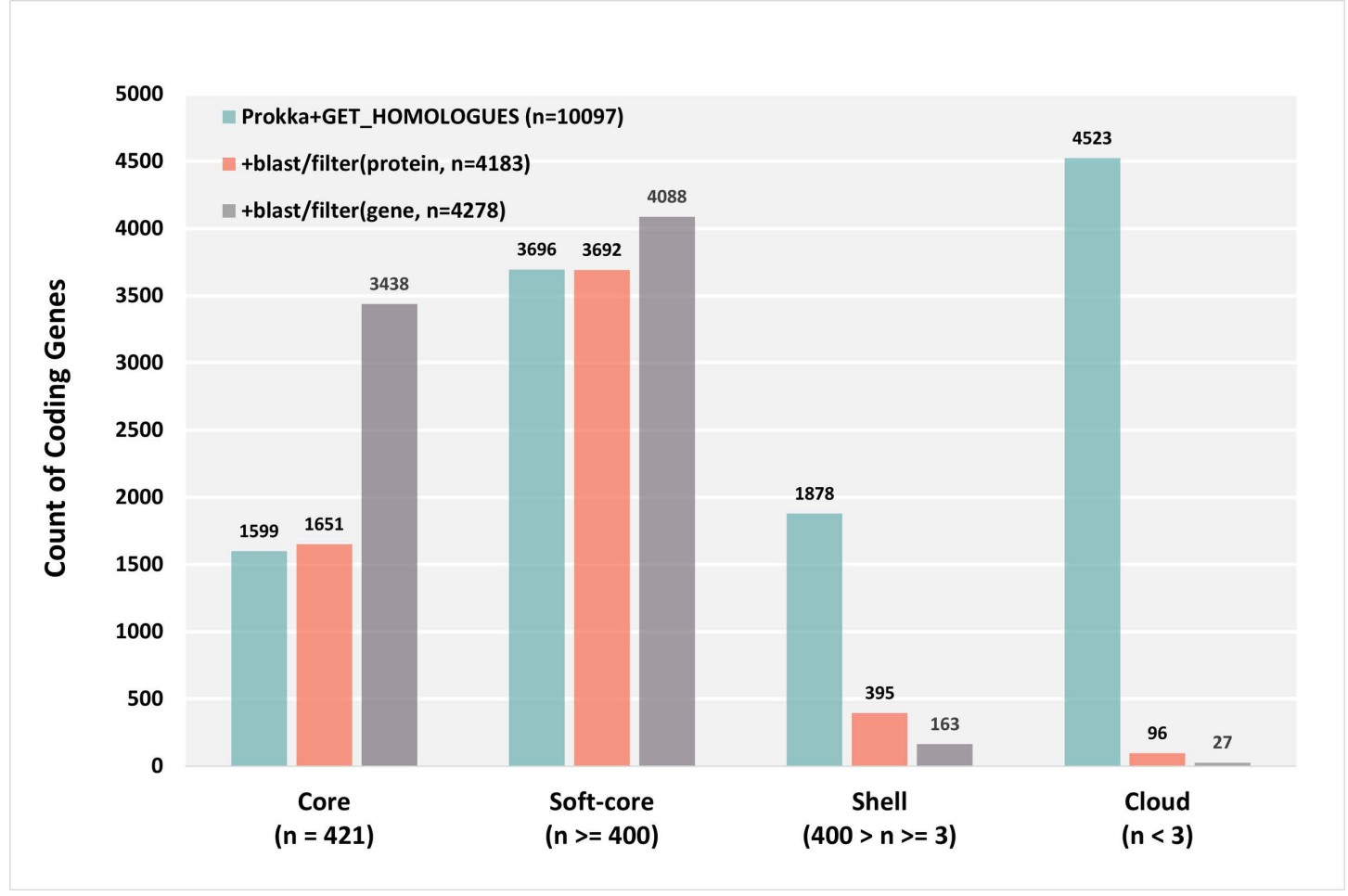

**Fig 1. Pangenome composition.** Pangenome composition at different levels and with different methods.

dataset; a "complex" variation of "indel" and "stop gained" was detected five times. 6,035 unique SVs only occur in single isolates in either coding genes or intergenic regions. In the 4,098 coding genes, 5,111 (69.69% of 7,334) only occur in single isolates. This level is comparable to the reported level of 47.5% only for small indels(1–40 bp) in the study of Coll.[53]

Based on the pangenome matrix, there are 99,694 interruptions detected in 420 strains affecting 2,447 coding genes. There were 23,221(23.29%) copies of 744 genes which could not be detected by *blastn*, of which 50.40% (11,704, including complex variations) could be attributed to large deletions detected resulting in the complete deletions of genes. Other SVs added another 46.26% (42,110) of the gene interruptions in coding genes which were not completely deleted. High impact SNPs alone caused 12.65% (12,615) interruptions. Small indels (< 100 bp) contributed the largest part of interruptions (31.22%) by a single category. Together, SVs including insertions of IS*6110* and high impact point mutation in start/stop codons classified 67,315 (67.52%) interruptions in the pangenome in total, which left 32.48% of the interruptions unclassified. (Table 1)

We further investigated the distribution of un-classified interruptions to identify possible causes. A total of 32,379 un-classified interruptions were observed in 1,328 genes. There were 253 genes having more than 21 (5% of total samples) un-classified interruptions. The un-classified interruptions in these 253 genes corresponded to 88.29% (28,587) of all the un-classified interruptions detected, indicating these genes were tend to produce low quality data with our pipeline. On the other hand, there were 33 (out of 421) samples each with > 50 un-classified interruptions of in total 3,635 (39.16%)

**Table 1. Genetic variations with interruptions in coding genes.**

| Category | Genetic Variation | Interruptions | % |
|---|---|---|---|
| SV | Indel | 31,123 | 31.22% |
| | Complete deletion | 11,682 | 11.72% |
| | Large deletion | 5,152 | 5.17% |
| | Large insertion | 1,463 | 1.47% |
| | IS*6110* | 1,735 | 1.74% |
| | complex SV | 2,659 | 2.67% |
| | **sub total** | **53,814** | **53.98%** |
| SNP | stop gained | 6,505 | 6.52% |
| | stop lost | 1,961 | 1.97% |
| | start lost | 4,148 | 4.16% |
| | complex SNP | 1 | 0.00% |
| | **sub total** | **12,615** | **12.65%** |
| **Complex** | | **886** | **0.89%** |
| **Interruption** | **Classified*** | **67,315** | **67.52%** |
| | **un-Classified** | **32,379** | **32.48%** |
| | **Total** | **99,694** | |

*: classified interruption means the interruptions in the gene are categorized by the variations detected by our pipeline, while un-classified interruption means no variation is detected in the corresponding genes interrupted.

un-classified interruptions, indicating these samples might have yielded lower sequencing quality and more errors in assembly, but this bias was less obvious as compared to the 253 genes with low quality mentioned above. After excluding the 253 low quality genes, there were 46,024 interruptions in 2,194 genes and only 8.24% (3,792) of the interruptions were not classified by genetic variations.

Some of the genetic variations were phylogenetically specific, such as TbD1 present only in Lineage 1, which harbored two ORFs *mmpS6* and *mmpL6*, and RvD2 absent in Lineage 2, Lineage 3 and H37Rv which harbored four ORFs including *pimC*, *suoX*, *15952_hypothetical_protein* and *15953_transmembrane_transp* (Fig 2 and S1 table). In addition, we have identified one LSP which was lineage specific but not reported before and one LSP only recently reported.[7] A fragment of 3,138 bp (inserted between 1,414,558-1,415,891 in H37Rv) deleted in Lineage 2 and Lineage 4 strains harbored two ORFs. This fragment of insertion sequence to H37Rv could be fully matched to other MTBC genomes in NCBI database but not to the H37Rv genome, possibly another LSP yet to be reported. The another LSP of 4,494 bp (inserted to H37Rv at 2,219,418) was (partially) deleted in all Lineage 4 strains but presented in all other genotypes, which contained six ORFs and has been reported in a previous study.[7]

On the other hand, some LSPs were found in highly polymorphic regions where multiple overlapping SVs were identified in different genomes. For example, RD3 (Fig 2, from *Rv1573* to *Rv1586c*), which harbored 15 genes (including *Rv1575* which was interrupted by indels or complete deletions in all other genomes except H37Rv), was deleted in Lineage 2 strains; however, RD3 was also deleted in 21 Lineage 4 strains. In the region where RD152 (Fig 2 from *Rv1754c* to *Rv1762c*) was located, multiple deletions were observed in Lineage 2 and Lineage 4 strains but with different borders: seven genes (including *cut1* of low quality) were absent in Lineage 2 strains due to the deletion from 1,986,638–1,998,622; five genes (including *cut1* of low quality) were deleted in 21 Lineage 4 strains due to the deletion from 1,987,702–1,998,657.

High impact SNPs or other types of SVs could also result in interruptions of genes in specific lineages. For example, *Rv0197* was interrupted by an insertion of 2 bp at position 234,496 in Lineage 2 strains but also in all other strains except H37Rv; *ephF* was interrupted by the deletion of one nucleotide at site 162,152 in Lineage 2 strains; *Rv0061c* was

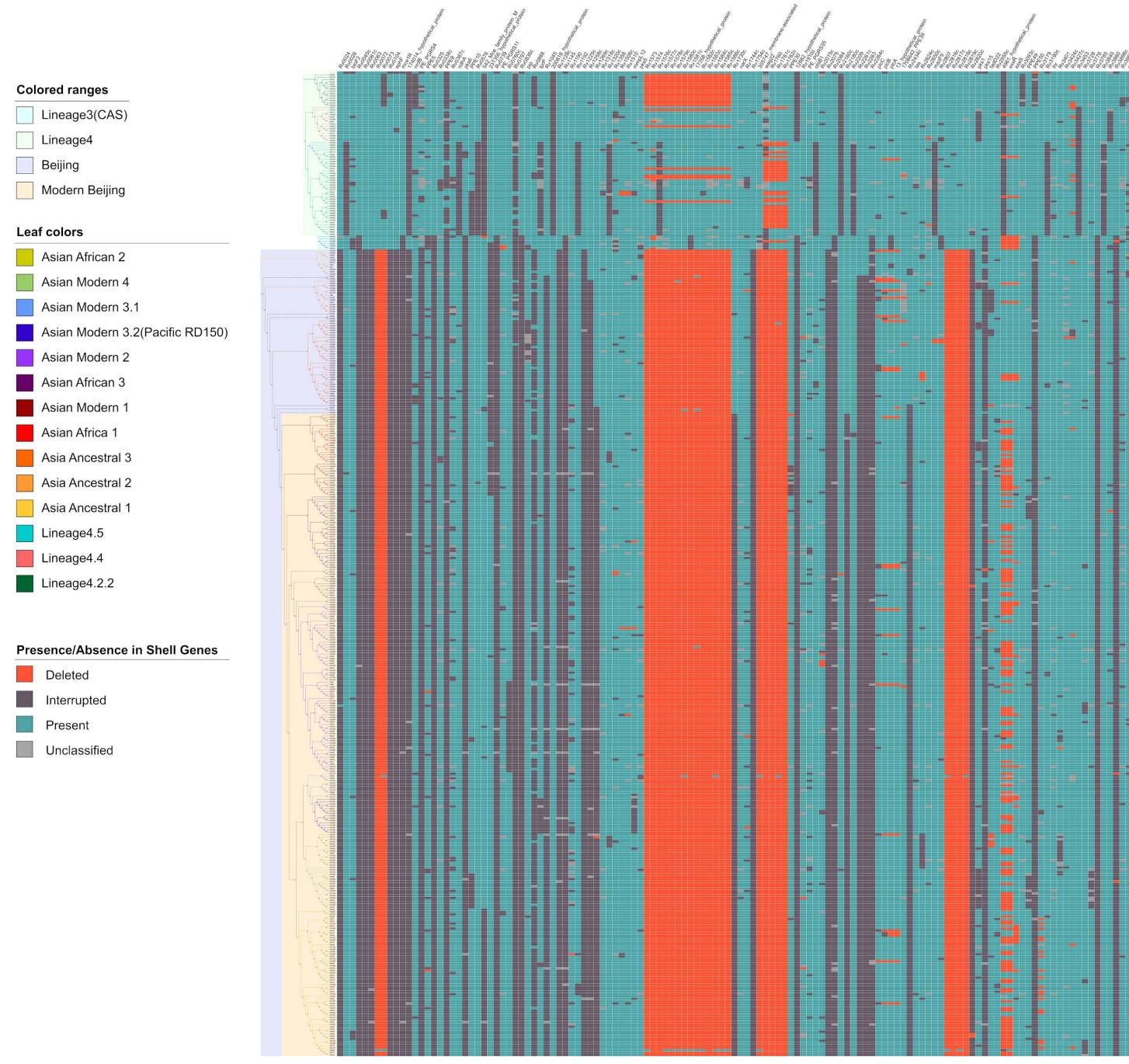

**Fig 2. Genetic variations in shell genes.** 128 shell genes in the pangenome with information of presence and absence at gene level and protein level in each sample. Detailed information about these 128 shell genes could be found in S1 table. Low quality genes and non-Rv genes were not shown. The phylogeny tree on the left side was constructed in our previous study using 33,220 SNPs detected in the whole genome with reference to H37Rv using max likelihood method.[20] The genetic distance information was removed for better presentation of the topology of the tree. Bright red boxes represent genes completely (>90% of length) deleted in that genome; dark grey boxes represent genes presenting in the genome but were interrupted by SVs or high impact SNPs; blue boxes represent genes with valid ORFs; light grey represent genes with un-classified interruptions.

interrupted by a point mutation from G to T at position 65,150 which resulted in the gain of an extra stop codon in Lineage 2 strains. *15427_dxs* was interrupted by IS*6110*–15 insertion in H37Rv; in our dataset, no other genomes had this IS*6110* insertion but at least 351 genomes had an insertion of 5 bp in this gene, so only 66 genomes (all Lineage 4) had valid ORFs of this gene in our dataset.

### Landscape of genetic diversity shaped by selective pressures, population structure and purifying selection

Many factors could shape the landscape of genetic diversity in MTB, such as selective pressure, population structure and purifying selection. To investigate how genes evolve under selective pressures, we calculated Tajima's D, nucleotide diversity π and detected homoplasy events in coding genes. The 253 coding genes that might have inferior sequencing quality and the 85 non-Rv genes that were prone to errors with short reads sequencing were excluded from this analysis, thus 3,845 genes were analyzed out of the total 4,183 coding genes.

Nucleotide diversity (π) measures the average number of nucleotide differences per site between any two DNA sequences chosen randomly from the population.[54] It's a measure of genetic variation within a population. Nucleotide diversity π was calculated for 3,818 coding genes with more than one CDSs of valid ORF which varied between 0 to 7.09e-3. The average nucleotide diversity was 1.51e-4 in this pangenome. Stratified by the value of Tajima's D, the values of π distributed almost symmetrically around the average. With reference to the average nucleotide diversity, there were

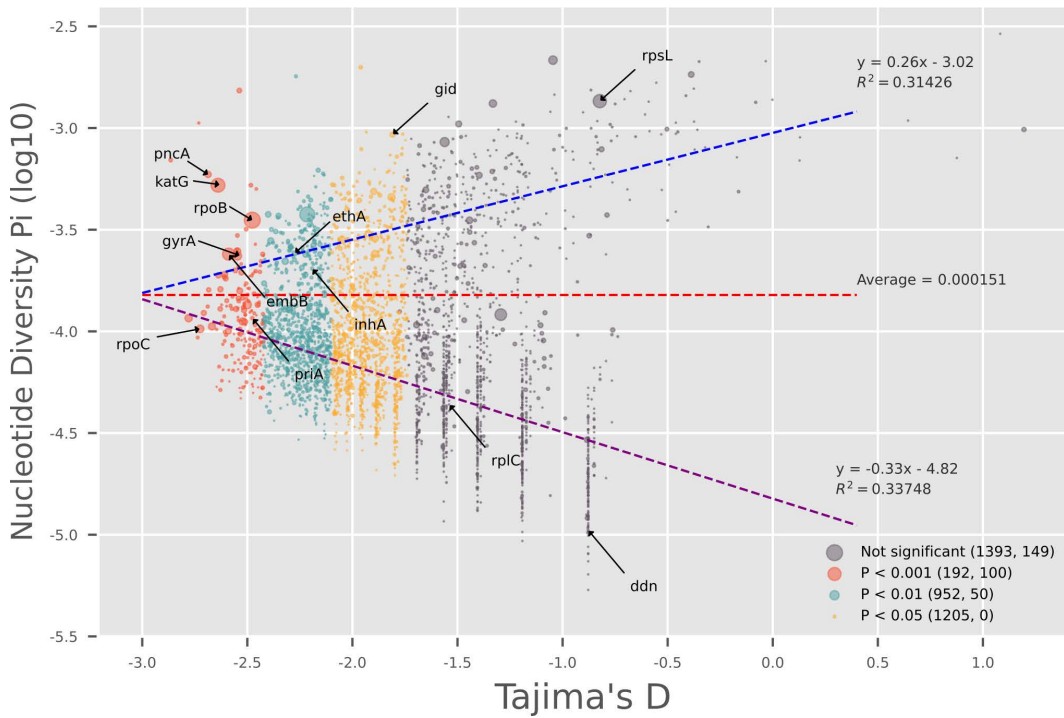

**Fig 3. Selective pressure, nucleotide diversity, and homoplasy in the pangenome (n = 3,742).** Only high quality Rv genes with > 3 CDSs of valid ORFs and at least one segregating nucleotide site are shown. Each circle represents a gene. x axis is Tajima's D, and y axis is nucleotide diversity π. Count of homoplasy events are shown by the size of the circles. The colors show the different levels of significance of Tajima's D. In the legend, the first number in parenthesis is the number of genes at that level of significance for Tajima's D, and the second number shows the corresponding number of counts of homoplasy events as shown by corresponding circle size. The red line shows the average nucleotide diversity across the pangenome. The blue line shows the predicted π by linear regression model for genes above the average nucleotide diversity, and the purple line shows the predicted π below the average nucleotide diversity.

957 (25.07%) genes of higher nucleotide diversity and 2,861 (74.93%) genes of lower nucleotide diversity, indicating that the nucleotide diversity was concentrated in a small proportion of coding genes (Fig 3).

Tajima's D is a statistical test used to compare the number of segregating sites to the average number of pairwise differences in a DNA sequence for a given sample set. A significantly negative value of Tajima's D indicates recent selective sweep, population expansion after a recent bottleneck, or linkage to a swept gene; a significant positive value of Tajima's D indicates balancing selection or sudden population contraction; and a zero value of Tajima's D indicates no selection taking place.[44] Tajima's D was obtained for 3,742 coding genes with > 3 valid ORFs and at least one segregating nucleotide site, 2,349 of which have shown significant signal of selective pressure (adjusted P-value < 0.05) and 192 showed strong signal of selective pressure (adjusted P-value < 0.001) (Fig 3). In our previous study [20], we have demonstrated that the same sample set in this study had experienced a population contraction recently. Thus, the observed significant negative value of Tajima's D suggests recent selective sweeps or directional evolution, and the existence of selective pressures rather than population expansion. Most of the drug resistance related genes including *gyrA*, *priA*, *rpoB*, *rpoC*, *katG*, *pncA*, and *embB*, showed signals of selective pressure, except *rpsL, rplC* and *ddn*. While *rplC* and *ddn* were conserved in sequences (π < 1e-5), *rpsL* showed high nucleotide diversity (π = 0.001356). The genetic variation in *rpsL* in our dataset was constrained to four nucleotide loci and mostly in codon 43 (72.92% of K, 0.24% of M, and 26.37% of R). On the contrary, in other drug resistance related genes under selective pressures, such as in *katG*, many low frequency alleles were observed (Fig 4). There were 71 genes with > 3 CDSs of valid ORFs having no segregating sites thus no genetic variation detected, 40 of which were core genes, suggesting these core genes may carry critical functions which bear no mutation at all.

Homoplasy occurs when a trait or nucleotide sequence is shared by two or more taxa due to convergent evolution, parallel evolution, or evolutionary reversals, rather than shared ancestry.[55] Because MTB has a very stable genome and no horizontal gene transfer, the possibility of parallel evolution and evolutionary reversals are estimated to be very low. In this case, homoplasy can be used as an index of positive selection (convergent evolution) due to survival advantages. Only 1,745 (19.91%) coding genes show signals of positive selection detected as homoplasy. We have identified 6,695 homoplasy events including 3,159 SVs/high impact SNPs homoplasy events and 3,536 homoplasy events with non-synonymous SNPs. There were 129 genes showing higher level of homoplasy (>= 9 homoplasy events). The most significant non-synonymous homoplasy SNPs in our dataset was detected in codon 315 in *katG*, which was estimated to have mutated 92 times (Fig 5). Interruptions of *esxR* were observed 82 times due to different deletions and insertions, which was the most frequently interrupted gene by SVs (Fig 2.)

MTB has a typical clonal population structure with no horizontal gene transfer which introduces systematic genetic variations between sub-populations known as population stratification.[56] Positive Fst values are positively correlated with the level of population stratification. In total, 1,267 coding genes had significant positive values of Fst (P-value < 0.05) based on SNP information in coding genes varying between 0.0004 to 1, indicating these genes were subject to the impact of population stratification at different levels. In addition, complete deletions or interruptions of genes in one sub-group would result in lineage-specific genes in other sub-groups, which is another type of systematic difference between genotypes and could be considered as population stratification. We have detected at least 70 lineage-specific genes absent in Lineage 2 or Lineage 4 in this sample set. In Lineage 2 strains, 40 genes were absent due to LSPs, including 14 phage related genes located around the RvD3 deletion and five CRISPR associated genes located in RD207.

**Over-represented gene categories following different evolutionary patterns**

As shown in Fig 2, the relationship between selective pressure and nucleotide diversity was not mono-linear but bilateral. For genes of higher nucleotide diversity than average, the nucleotide diversity was negatively correlated with the strength of selective pressures (or positively correlated with the value of Tajima's D, correlation coefficient = 0.26); for genes with low nucleotide diversity, the nucleotide diversity was positively correlated with the strength of selective pressure (correlation coefficient = -0.33) (Fig 3). When the data was stratified by the level of selective pressure and nucleotide diversity,

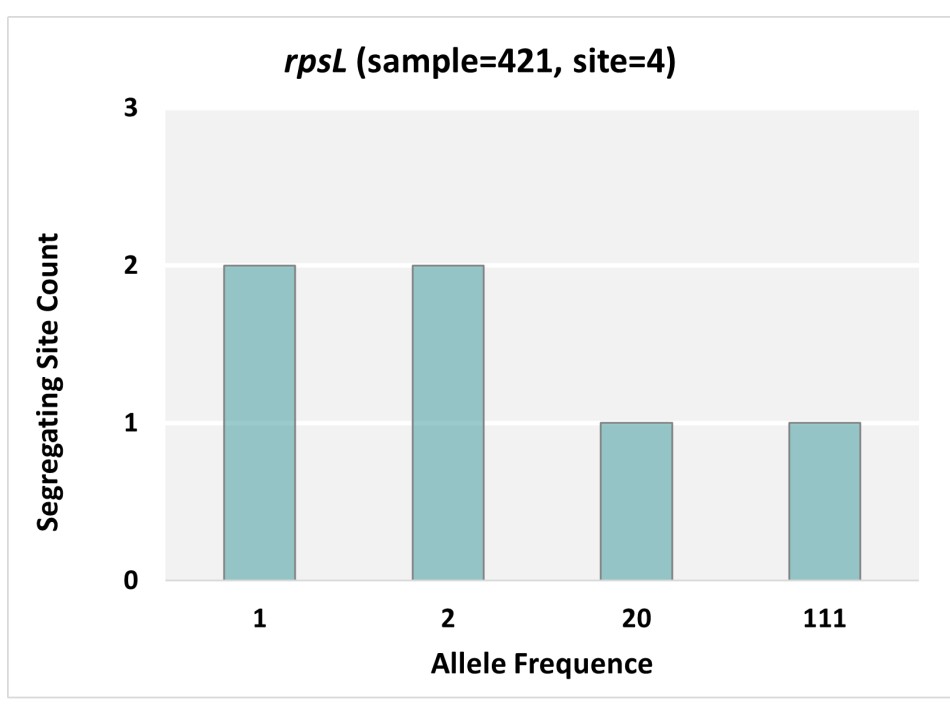

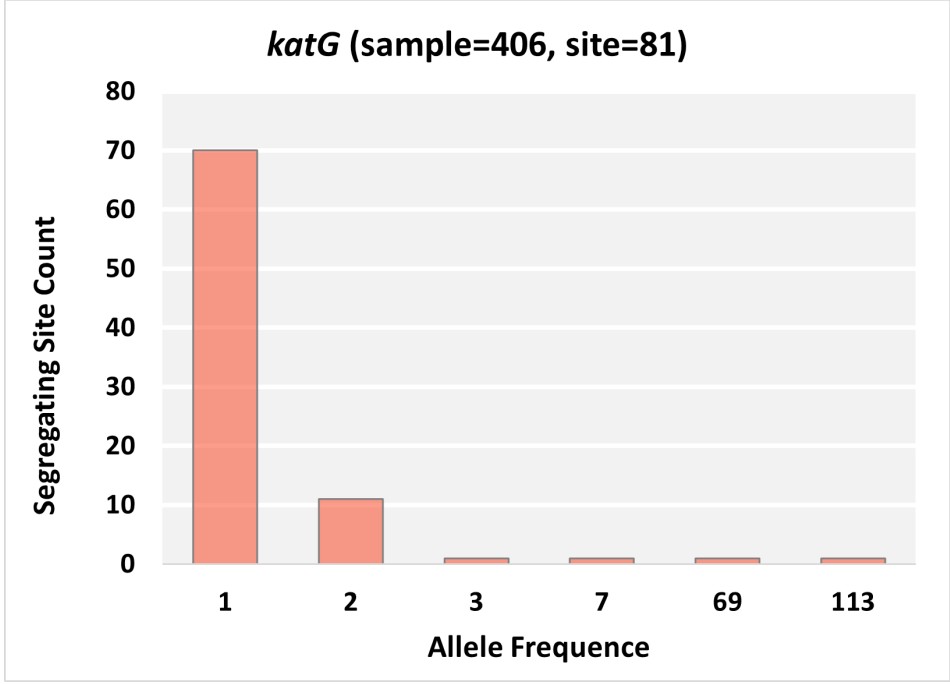

**Fig 4. Folded minor allele frequency spectrum for *rpsL* (a) and *katG* (b).**

these trends were confirmed by the average nucleotide diversity in each sub-group. For example, in the group of genes with higher nucleotide diversity than average, the average nucleotide within each subgroup increased almost mono-directionally as the selective strength decreased; while in the group of genes with lower nucleotide diversity than average, the trend was the opposite. (Table 2)

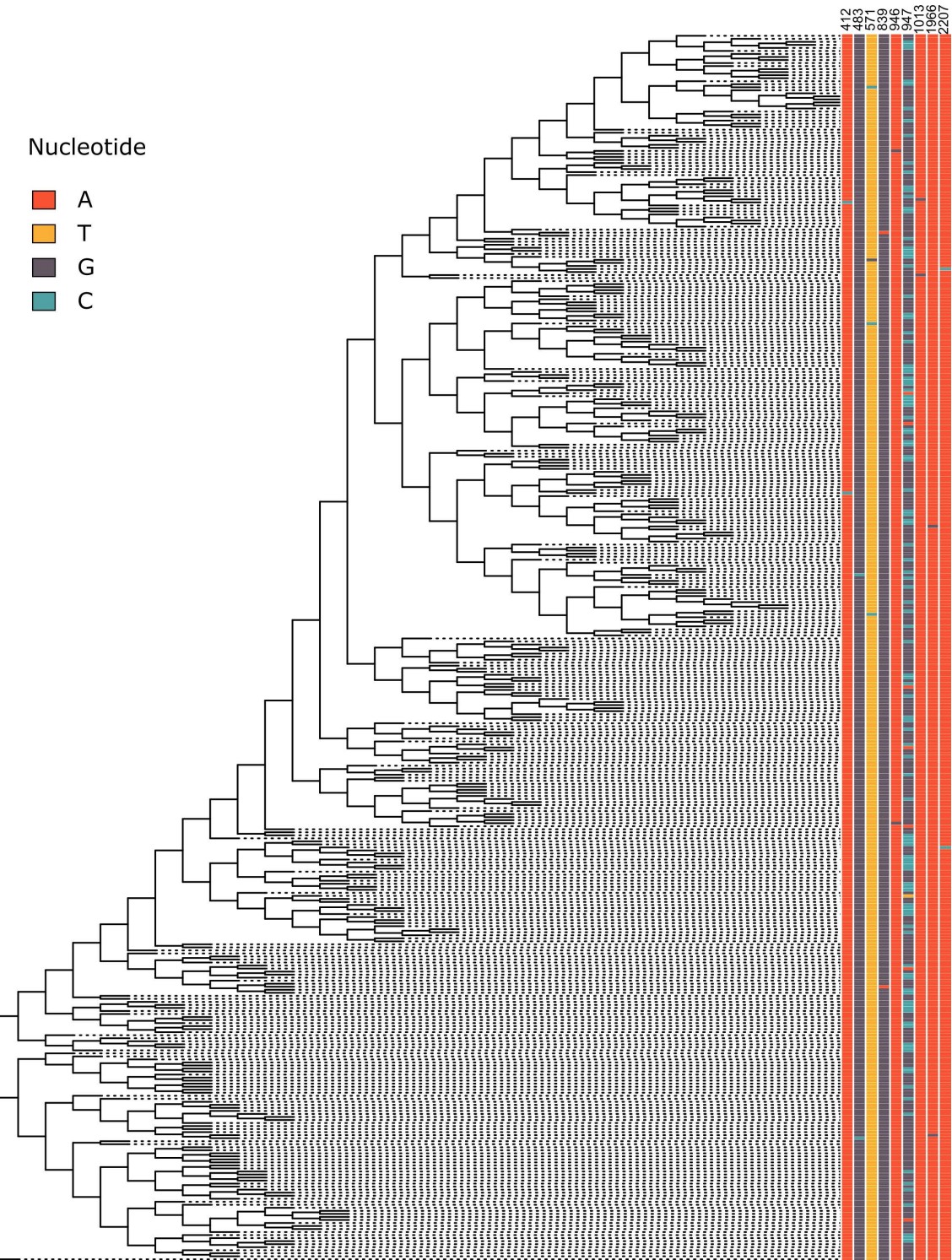

**Fig 5. Homoplastic SNPs in *katG*.** Phylogeny tree shows 404 genomes with valid *katG* ORFs. The phylogeny was modified from the same one used in **Fig 2** but with genomes of in valid *katG* ORFs removed. Nucleotide site 947 corresponds to 944 in H37Rv due to a 3 bp insertion in some genomes upstream of this position. Strips show the sites with homoplasy with nucleotide in different colors: A = red, C = blue, G = grey, T = orange.

**Table 2. Population genetic characteristics of genes following different evolutionary patterns.**

| Nucleotide Diversity | Selective Pressure# | Gene | Core Genes | Gene with Homoplasy | Average Fst | Average Homoplasy Events | Average Nucleotide Diversity |
|---|---|---|---|---|---|---|---|
| High | Strong | 45 | 6 | 43 | 0.46 | 15.29 | 2.94e-4 |
| | Moderate | 210 | 42 | 158 | 0.48 | 3.80 | 2.72e-4 |
| | Significant | 294 | 102 | 183 | 0.52 | 2.15 | 3.22e-4 |
| | Not Significant | 405 | 187 | 174 | 0.56 | 1.09 | 5.14e-4 |
| Low | Strong | 147 | 29 | 112 | 0.41 | 4.42 | 9.42e-5 |
| | Moderate | 742 | 251 | 452 | 0.31 | 1.77 | 8.31e-5 |
| | Significant | 911 | 448 | 362 | 0.21 | 1.04 | 7.26e-5 |
| | Not Significant | 988 | 546 | 232 | 0.10 | 0.69 | 5.07e-5 |
| | | 3,742* | 1,611* | 1,716* | 0.38$ | 3.88$ | 1.51e-4$ |

#: Different levels of selective pressures were defined by the significance of P-value of Tajima's D. Strong selective pressure was defined by P-value < 0.001; moderate selective pressure was defined by P-value < 0.01; significant selective pressure was defined by P-value < 0.05. Not significant was defined by P-value > 0.05.

*: Sum of all groups.

$: Average of all groups.

Under selective pressure (adjusted P-value for Tajima's D < 0.05), 1,310 genes have shown signals of positive selection as indicated by homoplasy. The level of positive selection was positively correlated with the strength of selective pressures irrelevant to nucleotide diversity, as the average number of homoplasy events detected increased as the strength of selective pressures became stronger in both the high and low nucleotide diversity groups. (Table 2) For example, among the 192 genes under strong selective pressure (adjusted P-value < 0.001), 155 (80.73%) genes showed homoplasy, and the average number of homoplasy events was 15.29 and 4.42 for high and low nucleotide diversity groups, respectively; among the 1,393 gene showing no signal of selective pressures, only 406 (29.15%) genes showed homoplasy, and the average number of homoplasy events was 1.09 and 0.69 for high and low nucleotide diversity groups, respectively. There were 35 genes showing high levels of homoplasy under strong selective pressure, suggesting on-going intensive adaptive evolution in these genes.

In the low nucleotide diversity group, when no signal of selective pressure was detected, 988 genes showed the lowest nucleotide diversity (average π = 5.07e-5), the lowest level of homoplasy (0.69 homoplasy events on average), and the lowest level of fixation between subpopulations (average Fst = 0.10), indicating extreme evolutionary conservation of these genes in MTB which were assumed to be subject to purifying selection. (Table 2) There were 500 core genes (at protein level) among these highly conserved genes with no positive selection or selective pressures signal detected. As mentioned above, there were 40 core genes showing no genetic variation which could not be calculated for Tajima's D and are not included in this table. Together this makes a total of 540 conserved core genes.

In the high nucleotide diversity group, 405 genes showing no signal of selective pressure had the highest nucleotide diversity (average π = 5.14e-4) and also the highest level of fixation between subpopulations (average Fst = 0.56) and a moderate level of homoplasy (1.09 homoplasy events on average), indicating that population stratification was the major contributor to the nucleotide diversity for this group and a relatively low level of conservativeness in evolution for these genes. (Table 2)

A gene's characteristics in the population genetics and presence in the pangenome can provide insights into the role of a gene in evolution. Based on the above described characteristics, we divided coding genes into five groups: 1) genes under selective pressures (n = 2,349) including the 192 genes under strong selective pressures and the 35 genes showing high levels of homoplasy under strong selective pressures; 2) the highly conserved core genes (n = 540) with

no homoplasy or selective pressures detected; 3) genes showing no signal of selective pressures but with homoplasy detected (n = 435); 4) genes only influenced by population stratification (n = 202) showing neither signal of selective pressures nor positive selection, including lineage-specific genes; and 5) the other genes (n = 319) in the pangenome which showed no signal of selective pressures, no homoplasy, and no population stratification. We further investigated the over-represented gene categories in some characteristic group/sub-groups. (Table 3)

Critical biological processes, such as regulation of growth (GO:0040008), gene expression (GO:0010467), and translation (GO:0006412) were over-represented in the group of conserved core genes. Genes encoding cellular components of organelles such as ribosome (GO:0005840), and genes with toxin sequestering activity (GO:0097351) were also over-represented in this group.

Genes involved in actinobacterium-type cell wall biogenesis (GO:0071766) were over-represented in the group of genes under strong selective pressures. Especially, genes of fatty acid synthase activity (GO:0004312) and of phosphopantetheine binding (GO:0031177) activity were over-represented in this group also potentially related to cell wall biogenesis (Fig 6a).

Disruption of host anatomical structure (GO:0141060) was over-represented by five genes in the group of high-level homoplasy genes, four of which had phospholipase activity (GO:0004620) including all three proteins of sphingomyelin phosphodiesterase activity (GO:0004767) reported so far (Fig 6b).

In the group of genes under strong selective pressures and showing high homoplasy levels, fatty acid synthase activity (GO:0004312) was over-represented by five genes in this group; phosphopantetheine binding (GO:0031177) activity was

**Table 3. Over-represented gene categories.**

| | GO item | Description | No. in H37Rv | Observed | Fold Enrichment | Adjusted P-value |
|---|---|---|---|---|---|---|
| **Strong selective pressures (n = 192)** | GO:0031177 | phosphopantetheine binding | 21 | 13 | 10.51 | 2.95E-09 |
| | GO:0072341 | modified amino acid binding | 22 | 13 | 10.03 | 6.85E-09 |
| | GO:0004386 | helicase activity | 24 | 10 | 7.07 | 1.27E-04 |
| | GO:0005524 | ATP binding | 356 | 41 | 1.96 | 2.40E-03 |
| | GO:0004312 | fatty acid synthase activity | 16 | 7 | 7.43 | 5.54E-03 |
| | GO:0016887 | ATP hydrolysis activity | 116 | 19 | 2.78 | 1.01E-02 |
| | GO:0008094 | ATP-dependent activity, acting on DNA | 32 | 9 | 4.78 | 2.07E-02 |
| | GO:0005694 | Chromosome | 11 | 5 | 7.04 | 2.09E-02 |
| | GO:0071766 | Actinobacterium-type cell wall biogenesis | 53 | 12 | 3.67 | 3.36E-02 |
| **High homoplasy (n = 129)** | GO:0141060 | disruption of anatomical structure in another organism | 7 | 5 | 22.22 | 3.80E-04 |
| | GO:0004767 | sphingomyelin phosphodiesterase activity | 3 | 3 | 31.10 | 1.97E-02 |
| | GO:0004620 | phospholipase activity | 13 | 5 | 11.96 | 2.00E-02 |
| | GO:0031177 | phosphopantetheine binding | 21 | 6 | 8.89 | 2.14E-02 |
| | GO:0072341 | modified amino acid binding | 22 | 6 | 8.48 | 2.87E-02 |
| **Conserved core (n = 540)** | GO:0006412 | Translation | 114 | 43 | 3.05 | 6.10E-10 |
| | GO:0050789 | regulation of biological process | 414 | 83 | 1.62 | 3.32E-04 |
| | GO:0040008 | regulation of growth | 71 | 23 | 2.62 | 4.24E-03 |
| | GO:0005840 | Ribosome | 63 | 33 | 4.69 | 1.94E-14 |
| | GO:0003676 | nucleic acid binding | 457 | 85 | 1.66 | 3.75E-05 |
| | GO:0097351 | toxin sequestering activity | 19 | 9 | 4.47 | 1.55E-02 |
| **Strong selective pressures + High homoplasy (n = 35)** | GO:0031177 | phosphopantetheine binding | 21 | 6 | 26.77 | 5.02E-06 |
| | GO:0072341 | modified amino acid binding | 22 | 6 | 25.56 | 6.85E-06 |
| | GO:0004312 | fatty acid synthase activity | 16 | 5 | 29.28 | 4.77E-05 |

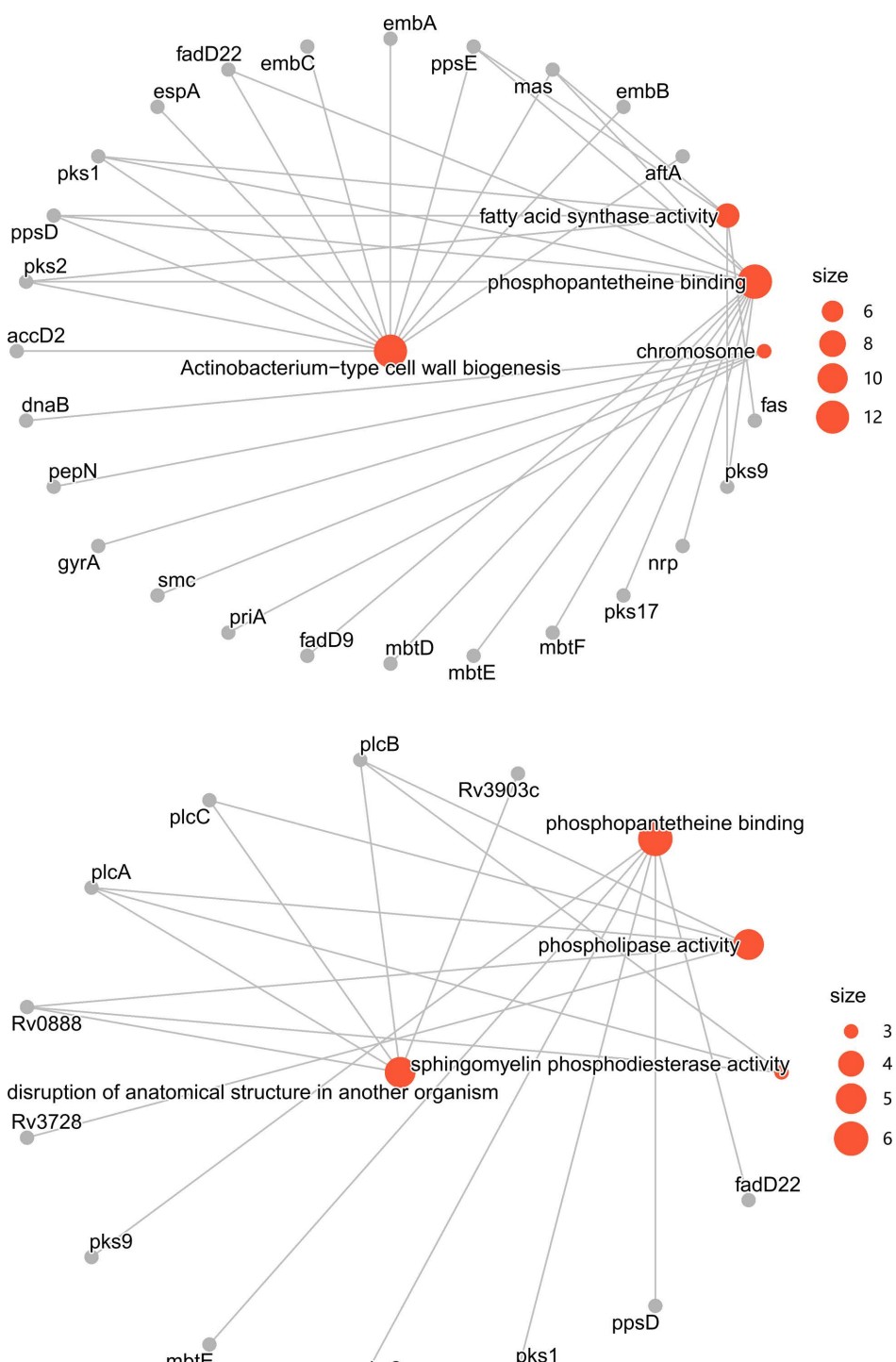

**Fig 6. Network of Enriched Ontologies. a)** Enriched ontologies in the group of genes under high selective pressures. Only genes enriched in actinobacterium-type cell wall biogenesis (GO:0071766) and related ontologies were shown. **b)** Enriched ontologies in the group of genes showing high homoplasy levels. Only genes enriched in disruption of anatomical structure in another organism (GO:0141060) and related ontologies were shown. Enriched ontology nodes were in red with gene nodes in grey.

over-represented by six genes in this group as well. Besides, drug resistance related genes were gathered in this group including *rpoB* (421 presences in the pangenome), *rpoC* (407), *katG* (406), *embB* (421), and *gyrA* (420).

## Discussion

Our study has shown that MTB has a highly conserved pangenome with only 85 new genes not presented in the reference genome H37Rv in a diversified sample set. The pangenome size of 4,278 estimnated here is very close to the value of 4,252 reported in a recent study by Behruznia *et al.*,.[57] This is expected as MTB has "conserved" genomes and a clonal population structure. By applying further filtering steps, the number of accessory genes was reduced almost by two thirds of the automatically predicted number. The inflated accessory gene number is mostly caused by alternative ORFs predicted by annotation software and platform-wide and workflow specific bias.[58,59] This result emphasizes the importance of establishing a standardized and optimized analytical process for pangenome analysis.

The size of pangenome of MTB is only one-fifth of *E. coli*, but the proportion of core genes is three times of that in *E. coli* at protein level and eight times at gene level.[60] This small pangenome size is presumably a result of the obligate intracellular lifestyle.[61] The average nucleotide diversity for coding genes is at least one order lower than that in other organisms.[62,63] These findings agree with and further confirm the notion that MTB is evolutionarily conserved as reported by previous studies using a standard reference genome or focusing on a specific region.[64–66]

Due to interruptions of ORFs caused by various types of genetic variation, about half of the core coding genes were potentially inactivated in different genomes. Structural variation accounts for more than half of the observed interruptions in coding genes while high impact SNPs only correspond to about ten percent, suggesting SVs are a major source of genetic diversity and is the major contributor to adaptive loss in evolution.[67] Although the number of unique SVs detected is only one fourth of the number of SNPs (7,334 v.s. 33,220) in the same sample set, the impact of SVs is more fundamental than SNPs, as SNPs in inactivated genes are assumed to have no impact on the biological functions. This bias may influence the analysis pipeline relying on SNP information, such as genome wide association analysis, which may require refinement of these methodologies.

Though genetic variations in LSPs are relatively limited in MTB, we have identified two genes, *pimC* and *suoX*, which are deleted due to the LSP of RvD2, may have crucial impact on metabolism. PimC is an amannosyltransferase involved in cell wall biosynthesis and some of its products are important virulence factors in MTB.[68,69] Searching literature, we have not found other homologues gene in MTB so far, we assume there is an alternative pathway involved in the biosynthesis of $Ac_nPIM_3$ in Lineage 2 which has not yet been discovered. *SuoX* oxidizes sulfite to sulfate directly in sulfur metabolism. Due to its nucleophilicity and strong reductive capacity, sulfite can be toxic for bacteria but can be detoxified by oxidation to sulfate.[70] Direct oxidation of sulfite seems to be the exclusive way to detoxify sulfite via oxidation, yet the gene *suoX* encoding sulfite oxidase in MTB is deleted in Lineage 2 strains and no other sulfite oxidase has been identified in MTB so far, which means the direct oxidation of sulfite in Lineage 2 may be compromised.[70] In the reverse direction, the assimilation pathway of sulfate, sulfite could be further reduced to sulfide, which is the form of sulfur required for biosynthesis of sulfur containing metabolites such as cysteine and acetate.[71] Sulfate assimilation is important for bacteria persistence. For example, $H_2S$ has been shown to stimulate respiration, growth and pathogenesis of MTB *in vivo*.[72] MTB can produce endogenous $H_2S$ from cysteine by a desulfhydrase (probably *Rv3684*).[73] The absence of *suoX* in Lineage 2 might contribute to the higher virulence of Beijing strains observed, as compromised oxidation capacity would increase the concentration of sulfite, which in turn moves the balance towards the direction of assimilation and consequently increases the production of $H_2S$. Besides *pimC* and *suoX,* the five CRISPR associated genes in the RD207 region deleted in Lineage 2 strains are reported to be associated with drug resistance.[74] These observations indicate that although LSPs only have a limited contribution to the difference in genetic content, some of these differences may have important consequences that their absences in certain genotype could result in differentiation in metabolism that might have contributed to the prevalence of certain genotypes.

Our data has shown that the intensity of positive selection is positively correlated to the strength of selective pressure in spite of other factors, indicating though it may be slow, the conservative MTB genome is still evolving. One important evolutionary force might be the interaction between MTB and the host immune system, as genes involved in disruption of host structures are over-represented in the group with a high level of homoplasy, indicating these selected mutations might help survive host immune attacks. Especially, the metabolism of fatty acids may have played a key role as indicated by the over-representation of genes with phospholipase activity and sphingomyelin phosphodiesterase activity. For example, *Rv0888*, an extracellular bifunctional enzyme with both nuclease and sphingomyelinase activities, can enhance the colonization ability of *Mycobacterium smegmatis* in the lungs of mice.[75] Deletion of *plcC*, the phospholipase C in MTB, was related to a persistent tuberculosis outbreak in the UK.[76] *plcC* was reported to be a hotspot of deletions in another study as well.[59] Further investigation of these genes may deepen our understanding of the pathogenicity of MTB.

A few exceptions were observed for the correlation between selective pressure and positive selection. One example is the *rpsL* gene, which shows a high level of homoplasy (104 homoplasy events) indicating positive selection of advantageous mutations, but no signal of selective pressure. Mutations in codon 43 in *rpsL* confer a high-level resistance to streptomycin. Streptomycin is the first anti-TB drug used in monotherapy of tuberculosis but has been replaced by oral drugs for almost two decades now. [77] The disappearance of signal of selective pressure reflects the changes in the standard treatment regime. Though this historical selective pressure on *rpsL* no longer exists the signal of positive selection could still be detected in our data. Another factor that might have contributed to the persistent signal of positive selection is fitness cost. Mutations in other drug resistance related genes, such as *rpoB*, usually cause different levels of fitness cost.[78] While strains with mutations in *rpsL* conferring streptomycin resistance are not affected by such deficiency.[79] Epistatis may also play a role in the high frequency of *rpsL* mutantions as shown in a study with *E.coli* which reported higher fitness in some double mutants of *rpoB/rpsL* than wild genotypes.[80] Also in our previous analysis using the same sample set, the association between drug resistance patterns and genotypes suggests that certain genotypes which developed resistance to streptomycin seem not to show any deficiency in further developing rifampin resistance.[81] However, this type of highly advantageous mutation without fitness cost in genes is very rare, as we only observed 32 genes in this category with no signal of selective pressures and a high level of homoplasy. Further checking the function of these genes might be helpful to identify historical selective pressures in the evolution of MTB. These observations suggest that the current landscape of genetic diversity in MTB is the outcome of complicated interactions between changing selective pressures, purifying selection, clonal population structure, and other factors such as fitness cost of mutations.

Our study has shown the necessity and advantages to switch from an approach using a standard reference genome focusing on SNPs and small indels to a more comprehensive pangenome approach with detailed genetic variation information. By combining pangenome, population genetics and gene ontology, new insights into the biology and the interaction of MTB with its environment may be gained. Future studies with more advanced sequencing techniques like long read sequencing and more versatile bioinformatic tools such as graph pangenome will remove the drawbacks in current methodology and improve our understanding about the epidemiology and pathogenicity of MTB.[82].

## Supporting information

**S1 Fig. Pangenome pipeline.**
(TIF)

**S1 Table. Information of 128 shell genes in Fig 2.**
(XLSX)

## Author contributions

**Conceptualization:** Yang Zhou, Richard Anthony, Dick van Soolingen.

**Data curation:** Shengfen Wang, Bing Zhao, Yuanyuan Song.

**Formal analysis:** Yang Zhou.

**Funding acquisition:** Yanlin Zhao, Dick van Soolingen.

**Methodology:** Yang Zhou.

**Project administration:** Hui Xia, Yanlin Zhao.

**Supervision:** Yanlin Zhao, Dick van Soolingen.

**Validation:** Xichao Ou, Yang Zheng.

**Visualization:** Yang Zhou.

**Writing – original draft:** Yang Zhou.

**Writing – review & editing:** Richard Anthony, Ping He, Dongxin Liu, Dick van Soolingen.

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
