## [Decision Letter · Decision Letter 0]

13 Jan 2025

PONE-D-24-51517Understanding the epidemiology and pathogenesis of Mycobacterium tuberculosis with non-redundant pangenomePLOS ONE

Dear Dr. Zhou,

Thank you for submitting your manuscript to PLOS ONE. After careful consideration, we feel that it has merit but does not fully meet PLOS ONE’s publication criteria as it currently stands. Therefore, we invite you to submit a revised version of the manuscript that addresses the points raised during the review process.

As can be seen in the reviews, the reviewers felt that this manuscript is a well-done study overall and contains valuable insights about the genomic structure and evolution of Mtb.  The reviewers noted some weakness that need to be addressed, including clarifying some of the terminology and methodology used, reconciling some of the numbers reported, and citing some additional relevant literature.

We look forward to receiving your revised manuscript.

Kind regards,

Thomas R. Ioerger

Academic Editor

PLOS ONE

“Authors received the funding

Ou Xichao & Zhao Yanlin

Grant numbers

Zhao Yanlin :  2022YRC2305203

Ou Xichao :  2022YRC2305204

Project Title & Grand Numbers: Establishment and application of China's AIDS and tuberculosis pathogen gene database and intelligent precision prevention and control platform (2022YRC2305200)

Sub-project Title & Grand Numbers : Creation of a national representative tuberculosis pathogen genetic sequence database and its analysis tools (2022YRC2305203)

Sub-project Title & Grand Numbers : Research and application of tuberculosis transmission network and its molecular analysis tools in China (2022YRC2305204)

Full name of funder

2022YRC2305203 & 2022YRC2305204 : Ministry of Science and Technology of the People's Republic of China

Url of funder

Ministry of Science and Technology of the People's Republic of China : https://www.most.gov.cn/index.html.”

3. Please note that your Data Availability Statement is currently missing a direct link to access each database]. If your manuscript is accepted for publication, you will be asked to provide these details on a very short timeline. We therefore suggest that you provide this information now, though we will not hold up the peer review process if you are unable.

Additional Editor Comments:

As can be seen in the reviews, the reviewers felt that this manuscript is a well-done study overall and contains valuable insights

about the genomic structure and evolution of Mtb. The reviewers noted some weakness that need to be addressed,

including clarifying some of the terminology and methodology used, reconciling some of the numbers reported, and citing some additional relevant literature.

Reviewers' comments:

Reviewer's Responses to Questions

**Comments to the Author**

1. Is the manuscript technically sound, and do the data support the conclusions?

Reviewer #1: No

Reviewer #2: Yes

2. Has the statistical analysis been performed appropriately and rigorously? 

Reviewer #1: I Don't Know

Reviewer #2: Yes

3. Have the authors made all data underlying the findings in their manuscript fully available?

Reviewer #1: Yes

Reviewer #2: No

4. Is the manuscript presented in an intelligible fashion and written in standard English?

Reviewer #1: No

Reviewer #2: Yes

5. Review Comments to the Author

Reviewer #1: The study by Zouh et al seeks to get new insight into the epidemiology and pathogenesis of Mtb strains by analyzing the pangenome, i.e. entire gene content of this species, which is usually neglected for this kind of studies (due to the clonal population structure). They address an important point that has been of major interest in the last few years. Overall the authors did a thorough analysis combining various tools to find a good consensus for annotations problems that can emerge in such a workflow. Findings about possibly disrupted ORFs, and hyperconserved genes I find very interesting. However, several parts are currently difficult to follow because the authors throw out various numbers that address different subsets, analysis steps, and the rationale and interpretation of those numbers is poorly presented. Below some comments that should be clarified/improved:

Major comments:

l.224 The pangenome is based on the population structure of Mtb strains in China. For instance lineage 3 (gene content) is completely missing. This limitation should be addressed in the title and the abstract.

l.40 and l 427 The statement that only like 50% of the core genome is intact and translated to functional proteins is difficult to follow from the results section. Usually, one allows some alleles to be missing or failing the thresholds in core genome schemes, so this number above seems just summing up disrupted ORFs in individual samples. If that is really an unusual finding, this needs to be presented better.

l. 227 this paragraph is difficult to understand. Where is the number of 1 million predicted CDSs coming from when on average 4 thousand genes are predicted per genome? And what does it mean they were identical to the blast results? The whole selection procedure of genes included for the pangenome may benefit from a flowchart.

l.285 I re-read this passage a couple of times, and could not grasp what the authors want to say here.

l.433 What is exactly the extend of structural variation as compared to SNPs in Mtb? You highlight RvD2 is that the known RD2 deletion in lineage 2? Any insides/observations on new (maybe phylogenetic) larger deletions?

l.457 Also like 60% of your core genome is under (positive?) selective pressure? Mtb is so far thought to be shaped by purifying selection.

Minor comments:

l.63 the term “ancient” vaccine is maybe a bit degrading

l.65 pretomanid might be mentioned as well then with regard to new drugs

l.70 introduce MTB abbreviation at first mentioning

l.83 Bottai et al https://www.nature.com/articles/s41467-020-14508-5 demonstrated the increased virulence of strains with a deleted TbD1 region

l.106 I would not say that nonsense or frame shifts are ignored in presence/absence gene content analysis. Usually that involves a de novo annotation, and current algorithms still can link genes that are truncated or have frameshift mutations with their homologue/functional counterpart.

l.123 explain selection criteria, what does it mean based on drug resistance patterns?

l.144 explain a bit why CDSs needed to be clustered, and what is meant with overlapping with Rv? That also affects the results part. How do you come to 10 thousand coding gene clusters?

l.198 briefly explain the TajimaD and pi measurements, and homoplasy.

l.214 what are “different” evolutionary patterns?

l.263 what is a valid ORF, and what is not valid?

l.266 what exactly do you mean with “how genetic variations affected the presence of absence at different levels in the pangenome”? do you mean by which genomic change certain genes were likely disrupted?

l.270 I assume the 76k sequence variants are detected in the whole dataset from a reference mapping approach. That does not include SNPs here right? Because the sentence above you are mentioning SNPs as well? And then you mention again “high-impact” point mutations. I would suggest to be very strict and consistent with naming of mutations and describe from which analysis which numbers come from.

L.357 rpsL high nucleotide diversity as compared to what?

Fig3 what is allele frequence here? And segregating site count? Allele = gene position? Or allele = gene sequence? Segregating site = SNPs?

Fig4 and 5 are not readable in the online version (low resolution)

Is Fig4 a phylogeny based on 127 shell genes? Which method? Seems to be a cladogram. In the caption it is stated it is based on 33k SNPs. In that regard one would rather use a ML or at least neighbor joining tree to have the genetic distances represented. Also the term shell gene is used for the first time here. Is that equivalent to the core genome here?

Reviewer #2: Comparative genomics is arguably the best tool for querying the Mtb’s evolutionary trajectory and the functional consequences thereof. In this manuscript, the authors presented a comprehensive computational pipeline to infer Mtb’s pan-genome architecture using 420 public short-read sequencing data of Mtb strains isolated through a single national survey conducted in China. The pipeline appears reasonable based on the methods described, though the code used for the analyses was not provided. The analysis revealed a closed pangenome comprising 4,278 genes. The authors reported an interesting finding that among the 4,098 annotated protein-coding genes, only 1,651 encode full-length proteins across all genomes, while the remaining genes were disrupted in at least a subset of strains. Furthermore, the authors reported known and potentially new phylogenetically structured structural variations (SVs) and large structural variations (LSVs) and estimated gene-wise selective pressures using a single metric.

Overall, the manuscript is well-structured and could be a valuable asset to the TB research community. I’d be happy to read this manuscript again in its published format if the authors could address the following minor concerns:

1. Which H37Rv genome assembly was used as the reference? Several H37Rv assemblies now exist, including curated assemblies generated using long-read sequencing (e.g., DOI: 10.7554/eLife.97870.1). Since the analyses were based on a single annotated H37Rv genome, the authors should confirm that they used a complete, high-quality genome assembly with no blind spots. They should also provide the specific assembly ID of the H37Rv reference genome used in this study.

2. I commend the authors for the incredibly large amount of work they did to improve the overall power and robustness in calling SVs and other genetic variations by integrating a handful of existing bioinformatic tools. That being said, this is not the first Mtb pan-genome paper and there were several recent studies that used long-read sequencing to generate complete genome assemblies and conduct SV analysis, which were not cited in the present manuscript. Please cite these relevant studies, such as the recent eLife paper by Behruznia et al. (DOI: 10.7554/eLife.97870.1) and the preprint by Marin et al. (DOI: 10.1101/2024.03.21.586149). These studies demonstrated that incomplete contig assemblies from short-read sequencing can lead to biases in accessory gene calling. Comparing findings with these datasets could help avoid spurious variant calls, particularly those discussed in lines 295-297.

3. The number of coding genes with interruptions varies between different parts of the manuscript. For example, line 40 states 2,447 interrupted genes, while line 298 reports 2,194. It could be me who mis-interpreted the words but I do think that these numbers would benefit from explicitly stating the criteria used for counting interrupted genes.

4. Terms such as "unclassified interruptions" (lines 285-295) and "complex" (Table 1) are unclear. Additionally, frameshift mutations are referenced in lines 180-181 but seem omitted from Table 1. Please provide precise definitions for terms like "unclassified interruptions" and "complex" mutations, and, if applicable, expand the legend for Table 1 to explain each mutation category.

5. The rationale for using Tajima’s D to infer selective pressure is unclear, especially when extensive SVs are expected to skew the metric toward extreme values. Please justify why Tajima’s D was chosen over alternative methods that may be more robust to structural variation, such as the McDonald-Kreitman test or dN/dS ratios.

6. Lines 315-316 report 40 genes absent in lineage 2 strains, many of which are phage-related or CRISPR-associated. A recent study (DOI: 10.1128/spectrum.00527-24) identified an association between drug resistance and the absence of CRISPR genes. The authors may consider comparing the results to this published study and testing for convergence in findings.

7. The homoplastic SVs near the plcC locus are intriguing and have also been identified in the Marin et al. preprint. Please cite this relevant preprint and perhaps also discuss potential biological implications of homoplastic SVs in this genomic region.

8. Fig. 5 lacks a legend explaining the color notations, and Figure 4 was placed before Figure 1 in the manuscript. The authors should also consider including a supplementary data sheet corresponding to Figure 4, which would greatly benefit readers who want to perform an in-depth re-analysis of this information-rich dataset.

6. PLOS authors have the option to publish the peer review history of their article (what does this mean?). If published, this will include your full peer review and any attached files.

Reviewer #1: No

Reviewer #2: **Yes: **Junhao Zhu

---

## [Author Response · Author response to Decision Letter 1]

27 Feb 2025

Done.

“Authors received the funding

Ou Xichao & Zhao Yanlin

Grant numbers

Zhao Yanlin : 2022YRC2305203

Ou Xichao : 2022YRC2305204

Project Title & Grand Numbers: Establishment and application of China's AIDS and tuberculosis pathogen gene database and intelligent precision prevention and control platform (2022YRC2305200)

Sub-project Title & Grand Numbers : Creation of a national representative tuberculosis pathogen genetic sequence database and its analysis tools (2022YRC2305203)

Sub-project Title & Grand Numbers : Research and application of tuberculosis transmission network and its molecular analysis tools in China (2022YRC2305204)

Full name of funder

2022YRC2305203 & 2022YRC2305204 : Ministry of Science and Technology of the People's Republic of China

Url of funder

Ministry of Science and Technology of the People's Republic of China : https://www.most.gov.cn/index.html.”

We have added the sentence “The funders had no role in study design, data collection and analysis, decision to publish, or preparation of the manuscript.” to the funding information.

3. Please note that your Data Availability Statement is currently missing a direct link to access each database]. If your manuscript is accepted for publication, you will be asked to provide these details on a very short timeline. We therefore suggest that you provide this information now, though we will not hold up the peer review process if you are unable.

We have added the link to the data analyzed to the manuscript.

We have re-written these parts in the manuscript to follow the requirements.

Responses to reviewer 1:

l.224 The pangenome is based on the population structure of Mtb strains in China. For instance lineage 3 (gene content) is completely missing. This limitation should be addressed in the title and the abstract.

Thank you very much for your comment. We have made the modification accordingly in the title and abstract.

l.40 and l 427 The statement that only like 50% of the core genome is intact and translated to functional proteins is difficult to follow from the results section. Usually, one allows some alleles to be missing or failing the thresholds in core genome schemes, so this number above seems just summing up disrupted ORFs in individual samples. If that is really an unusual finding, this needs to be presented better.

We refer to core genes those present in all 420 strains + H37Rv (n=421). If some gene is missing in less than 5% of the sample population (400 <= n < 421), we call them soft-core genes. Based on this definition, there were 3,438 core genes including some core genes which might harbor SVs and other variations. If there is no SV or other high impact point mutations, such as frameshift mutations/non-sense mutations, detected in any strain for a core gene, we consider it as intact and translated in this pangenome. Thus, it’s not the sum of the disrupted ORFs, which is at the individual coding sequence level; the concept of 50% intact core genes is at strain level.

l. 227 this paragraph is difficult to understand. Where is the number of 1 million predicted CDSs coming from when on average 4 thousand genes are predicted per genome? And what does it mean they were identical to the blast results? The whole selection procedure of genes included for the pangenome may benefit from a flowchart.

The 1 million predicted CDSs is the sum of the number of annotated CDS in all strains. Each strain has been annotated by Prokka of about 4000 CDSs. There are 420 strains. Thus, in total about 1,680,000 CDSs.

By “identical to blast results”, we refer to the comparison between the annotated CDSs by Prokka and blastn results using Rv gene sequences as query sequences and assemblies as subject sequences. If these two results have identical sequences, we consider the annotated CDSs identical to blast results.

We have produced a flowchart (Supplementary Fig 1.) to illustrate the procedure of producing the pangenome and data analysis to make this approach clearer.

l.285 I re-read this passage a couple of times, and could not grasp what the authors want to say here.

In this paragraph we investigated the possible causes for the unclassified interruptions. Because 32.48% of the interruptions cannot be matched to detected variations, which is not a small proportion, we want to check whether it’s due to the flaw of our pipeline or other technical reasons.

After further looking into the data and literature, we found that about 71% of the unclassified interruptions are probably due to the characteristics of the raw data produced on Illumina platform for MTB genomes, which might be introduced by high GC content/repetitive sequences and the limitations of the platform; in the remaining unclassified interruptions (30% of the total), about 39% were concentrated in 33 (out of 420) strains, indicating these 33 strains have more un-classified interruptions than other strains which might be due to low quality in the sequencing data/assembly.

After excluding these two factors, limitations of sequencing platform and low-quality samples, only 8% (3,792/32,379) of the unclassified interruptions remain. This observation suggests that other sequencing platforms, such as PacBio or NanoPore, could improve the quality of pangenome dramatically.

l.433 What is exactly the extend of structural variation as compared to SNPs in Mtb? You highlight RvD2 is that the known RD2 deletion in lineage 2? Any insides/observations on new (maybe phylogenetic) larger deletions?

With the same sample set of 420 MTB strains, we have detected high confidence SNPs at 33,220 unique positions with reference to the H37Rv genome (NC_000962.3) using current standard pipeline (original citation 20). There are 76,671 SVs detected in 4098 Rv coding genes in the same sample set, corresponding to 7,334 unique SVs (besides IS6110 insertions), but only 53,814 resulted in inactivation of coding genes in a subset of strains.

The RvD2 (length ~ 6,800 bp) is deleted in H37Rv due to IS6110 insertions at position 1987702. RD2 (length = 10,787 bp) is deleted in BCG but presents in H37Rv genome from 2,221,064 to 2,221,064, which was not detected in this sample set. See original ref 7 for details.

We have discovered one recently reported LSP (4,494 bp, inserted to H37Rv at 2,219,418, line 306-308) and one new LSP (3,138 bp, inserted between 1,414,558-1,415,891 in H37Rv, line 303-306) first reported in this study which are lineage specific and with annotated CDSs.

l.457 Also like 60% of your core genome is under (positive?) selective pressure? Mtb is so far thought to be shaped by purifying selection.

Because we only calculated Tajima’s for Rv genes with high quality data (excluding 253 low quality genes), and DNAsp requires at least 4 sequences (with valid ORFs) to calculate Tajima’s D, only 3,742 coding genes obtained the value of Tajima’s D.

2,349 (62.77%) out of these 3,742 coding genes showed signal of selective pressures, thus about 60% of coding genes are under selective pressure, among which 878 were core genes. (Table 2). We measured positive selection by detecting homoplasy events. Among the 878 core genes under selective pressures, 292 genes have been detected of homoplasy SVs or SNPs indicating positive selection.

This conclusion is also supported by another study using dN/dS (original citation 66). This can be expected as MTB is in an environment facing constant challenges such as host immune attack and nutrient deficit.

Purifying selection certainly plays a big role in MTB’s evolution, but the impacts of purifying selection are not even among coding genes, as 540 out of 1,651 core genes (at protein level) are highly conserved (no variation or lower nucleotide diversity than average) in our sample set and 2,861 (~67%) genes have lower nucleotide diversity than the average level, while more than 1,000 genes show signal of positive selection as suggested by homoplasy.

Minor comments:

l.63 the term “ancient” vaccine is maybe a bit degrading

BCG vaccine was first used medically in 1921, only 25 years after the first vaccine developed in history and which is more than 100 years now, so we agree ancient is hardly justified we now refer to it as an “old” vaccine.

l.65 pretomanid might be mentioned as well then with regard to new drugs

Thank you for your suggestion. We have added this to the introduction.

l.70 introduce MTB abbreviation at first mentioning

Done.

l.83 Bottai et al https://www.nature.com/articles/s41467-020-14508-5 demonstrated the increased virulence of strains with a deleted TbD1 region

Thank you for your suggestions.

l.106 I would not say that nonsense or frame shifts are ignored in presence/absence gene content analysis. Usually that involves a de novo annotation, and current algorithms still can link genes that are truncated or have frameshift mutations with their homologue/functional counterpart.

We agree that the statement is confusing. We have changed this part to make it precise.

l.123 explain selection criteria, what does it mean based on drug resistance patterns?

We have added this information from lines 124 to 128.

l.144 explain a bit why CDSs needed to be clustered, and what is meant with overlapping with Rv? That also affects the results part. How do you come to 10 thousand coding gene clusters?

To find new genes not in the reference genome (H37Rv), each strain needs to be annotated to identify CDSs. However, annotation software will sometimes assign the same CDSs in different genomes to different names or choose alternative ORFs for the same CDSs. So, annotation for different genomes will not automatically match each other, but need be compared by their sequence in a pair-wise manner to find their matches in each genome. This step is called clustering. Algorithms developed for this process include COG and OMCL.

As mentioned above, sometimes alternative ORFs are chosen by annotation software; in other cases, interruptions or variations will result in fusing or splitting of ORFs. Some of these ‘new’ ORFs are transcribed/translated, but in most cases according to current understanding (though we don’t know to what degree for present), they are not. But clustering algorithms will treat them as completely new genes. This will result in an inflation of the number of accessory genes. That is why after clustering step, there are more than 10k CDSs in the pangenome. For this reason, we introduced the blastn step to remove this artifact.

Blastn step will find the best hit of each Rv gene in other genomes. According to this study (original citation 27), alternative ORFs overlapping > 60 bp with the “true” ORFs are mostly falsely called. So, if two ORFs annotated in different genomes found to be overlapping > 60 bp in one genome by blastn, one of them is discarded and the Rv gene was kept in this study.

l.198 briefly explain the TajimaD and pi measurements, and homoplasy.

We have added the description in the results section.

l.214 what are “different” evolutionary patterns?

Our data shows that purifying selection, positive selection and various selective pressures all play important roles in shaping the population structure and current landscape of genetic diversity of MTB. However, these factors demonstrate their impact at different levels in different genes.

Coding genes could be divided into five groups: 1) genes under selective pressures (n=2,349) including the 192 genes under strong selective pressures and the 35 genes showing high levels of homoplasy under strong selective pressures; 2) the highly conserved core genes (n=540) with no homoplasy or selective pressures detected; 3) genes showing no signal of selective pressures but with homoplasy detected (n=435); 4) genes only influenced by population stratification (n=202) showing neither signal of selective pressures nor positive selection, including lineage-specific genes; and 5) the other genes (n=319) in the pangenome which showed no signal of selective pressures, no homoplasy, and no population stratification. However, these four classes are not strictly mutually exclusive, as all these evolutionary forces impact every gene at different levels in different classes which resulted in the continuous and complex landscape of genetic diversity in MTB.

We have added this explanation to the results section “Over-represented gene categories following different evolutionary patterns” from line 348 to line 417 to make this point clearer.

l.263 what is a valid ORF, and what is not valid?

By “valid ORF”, we refer to sequences of coding genes that could be translated into protein sequences by in silico method such as Biopython. We have added this explanation to the method (line 180) to make it clear.

l.266 what exactly do you mean with “how genetic variations affected the presence of absence at different levels in the pangenome”? do you mean by which genomic change certain genes were likely disrupted?

By “different levels”, we mean at gene level and at protein level. Complete deletions of genes would remove this gene sequence from genomes, while other variations, such as frameshift mutations, would keep (part of) this gene sequence in the genome but result in no protein product. Thus, different genetic variations would result in different variations at different levels in the pangenome.

l.270 I assume the 76k sequence variants are detected in the whole dataset from a reference mapping approach. That does not include SNPs here right? Because the sentence above you are mentioning SNPs as well? And then you mention again “high-impact” point mutations. I would suggest to be very strict and consistent with naming of mutations and describe from which analysis which numbers come from.1

Yes. The 76k variations don’t include SNPs.

The high impact point mutations are in the SNP category other than SV.

We have modified the sentence from line 274 to 280 to make it clear.

L.357 rpsL high nucleotide diversity as compared to what?

The nucleotide diversity of rpsL was compared to the average (=1.51e-4) and rplC and ddn. rplC and ddn also show no signal of selective pressure and are related to drug resistance like rpsL, but these two genes show lower nucleotide diversity than rpsL.

Fig3 what is allele frequence here? And segregating site count? Allele = gene position? Or allele = gene sequence? Segregating site = SNPs?

Allele frequency refers to the frequency of different alleles in the sample population.

Allele = variant; segregating site = polymorphic positions.

Fig4 and 5 are not readable in the online version (low resolution)

We have uploaded separate files for figures in tiff format which should meet the requirements for online publication.

---

## [Decision Letter · Decision Letter 1]

12 Apr 2025

PONE-D-24-51517R1Understanding the epidemiology and pathogenesis of Mycobacterium tuberculosis with non-redundant pangenome of epidemic strains in ChinaPLOS ONE

Dear Dr. Zhou,

Thank you for submitting your manuscript to PLOS ONE. After careful consideration, we feel that it has merit but does not fully meet PLOS ONE’s publication criteria as it currently stands. Therefore, we invite you to submit a revised version of the manuscript that addresses the points raised during the review process.

 You have appropriately addressed all of the reviewers' original concerns.However, there is one additional minor suggestion from one of the reviewers,

for which I would like to give you the opportunity to consider making a final revision.

We look forward to receiving your revised manuscript.

Kind regards,

Thomas R. Ioerger

Academic Editor

PLOS ONE

Journal Requirements:

Reviewers' comments:

Reviewer's Responses to Questions

**Comments to the Author**

1. If the authors have adequately addressed your comments raised in a previous round of review and you feel that this manuscript is now acceptable for publication, you may indicate that here to bypass the “Comments to the Author” section, enter your conflict of interest statement in the “Confidential to Editor” section, and submit your "Accept" recommendation.

Reviewer #1: (No Response)

Reviewer #2: All comments have been addressed

2. Is the manuscript technically sound, and do the data support the conclusions?

Reviewer #1: Yes

Reviewer #2: Yes

3. Has the statistical analysis been performed appropriately and rigorously? 

Reviewer #1: Yes

Reviewer #2: Yes

4. Have the authors made all data underlying the findings in their manuscript fully available?

Reviewer #1: Yes

Reviewer #2: Yes

5. Is the manuscript presented in an intelligible fashion and written in standard English?

Reviewer #1: Yes

Reviewer #2: Yes

6. Review Comments to the Author

Reviewer #1: Thanks for the clarification and improvements.

I found this statement in the abstract still a bit misleading:

"However, due to 99,694 interruptions in 2,447 coding genes, only 1,651 may be

translated in all samples, which dramatically reduces the number of active core genes."

(1) How many of those interruptions actually occur only in single isolates? Your core genome definition of 100% for >400 isolates is quite strict already. If you now count every isolate with a gene interruption, there are probably many singletons and possibly also artefacts included. When you use your softcore genome definition instead (or vice versa genes interrupted in >5% of all isolates), it would be more relevant and may hint to genes or sub-lineages which really make a phenotypic difference. Also we have no indication if those interruptions really abrogate the gene function (for this statement one should maybe consider only large InDels)

(2) In that regard we also saw that different tools miss or overinterpret structural variants. Visual inspection of a reference mapping helped to gain confidence for certain applications, and also BWA/GATK pipelines would identify at least small indels and confirm the results from the assembly used here.

Reviewer #2: The authors have addressed all of my concerns and kindly clarified the points I had previously misunderstood. I commend them for their excellent work and look forward to reading and sharing the manuscript in its published form.

7. PLOS authors have the option to publish the peer review history of their article (what does this mean?). If published, this will include your full peer review and any attached files.

Reviewer #1: No

Reviewer #2: No

---

## [Author Response · Author response to Decision Letter 2]

16 Apr 2025

Reviewer #1: Thanks for the clarification and improvements.

I found this statement in the abstract still a bit misleading:

"However, due to 99,694 interruptions in 2,447 coding genes, only 1,651 may be

translated in all samples, which dramatically reduces the number of active core genes."

(1) How many of those interruptions actually occur only in single isolates? Your core genome definition of 100% for >400 isolates is quite strict already. If you now count every isolate with a gene interruption, there are probably many singletons and possibly also artefacts included. When you use your softcore genome definition instead (or vice versa genes interrupted in >5% of all isolates), it would be more relevant and may hint to genes or sub-lineages which really make a phenotypic difference. Also we have no indication if those interruptions really abrogate the gene function (for this statement one should maybe consider only large InDels)

(2) In that regard we also saw that different tools miss or overinterpret structural variants. Visual inspection of a reference mapping helped to gain confidence for certain applications, and also BWA/GATK pipelines would identify at least small indels and confirm the results from the assembly used here.

(1) 6,035 SVs only occur in single isolates in either coding genes or intergenic regions. In the 4,098 coding genes, 5,111 (69.69% of 7,334) only occur in single isolates. This level is comparable to the reported level of 47.5% only for small indels(1-40bp) in the study of Coll (Coll F, Preston M, Guerra-Assunção JA, Hill-Cawthorn G, Harris D, Perdigão J, et al. PolyTB: A genomic variation map for Mycobacterium tuberculosis. Tuberculosis. 2014;94(3):346–54.) We have added this information to the result section line 287.

Indeed, the reviewer is correct these 99,694 interruptions likely include many artifacts. These artifacts are assumed to be concentrated in the 253 genes prone to platform-wide bias as stated in the results. Following the reviewer's suggestion we have revised the text to make this clearer (line numbers 24-28 document with revisions): "However, due to 99,694 interruptions in 2,447 coding genes, we can only confidently confirm 1,651 of these genes are translated in all samples. Assuming a proportion of these interruptions are artifacts, the number of active core genes would still be much lower than 3,483."

(2) There is currently no pipeline which can detect SVs as efficiently as SNPs. However, this study (Kosugi S, Momozawa Y, Liu X, Terao C, Kubo M, Kamatani Y. Comprehensive evaluation of structural variation detection algorithms for whole genome sequencing. Genome Biology. 2019 Jun 3;20(1):117.) suggests that some tools show better performance than others and overlapping results from different algorithms will improve accuracy. We have chosen tools according to our experience and following this strategy, used both tools working with assembly (e.g., minimap) and alignment.

In addition, we found for small indels (< 70 bp), several tools have shown high consistency with each other (> 90%), especially for bcftools which has shown high confidence by cross-checking with other tools. For large deletions (e.g, > 1000 bp), when there is a discrepancy between SV tools and presence/absence matrix, we visualized the region with IgV to confirm it.

---

## [Editor Report · Decision Letter 2]

22 Apr 2025

Understanding the epidemiology and pathogenesis of Mycobacterium tuberculosis with non-redundant pangenome of epidemic strains in China

PONE-D-24-51517R2

Dear Dr. Zhou,

We’re pleased to inform you that your manuscript has been judged scientifically suitable for publication and will be formally accepted for publication once it meets all outstanding technical requirements.

Kind regards,

Thomas R. Ioerger

Academic Editor

PLOS ONE
---

## [Editor Report · Acceptance letter]

PONE-D-24-51517R2

PLOS ONE

Dear Dr. Zhou,

I'm pleased to inform you that your manuscript has been deemed suitable for publication in PLOS ONE. Congratulations! Your manuscript is now being handed over to our production team.

Kind regards,

on behalf of

Dr. Thomas R. Ioerger

Academic Editor

PLOS ONE